# The Effect of Containment Measures during the Covid-19 Pandemic to Sedentary Behavior of Thai Adults: Evidence from Thailand’s Surveillance on Physical Activity 2019–2020

**DOI:** 10.3390/ijerph18094467

**Published:** 2021-04-22

**Authors:** Piyawat Katewongsa, Danusorn Potharin, Niramon Rasri, Rungrat Palakai, Dyah Anantalia Widyastari

**Affiliations:** 1Institute for Population and Social Research, Mahidol University, Salaya, Phutthamothon, Nakhon Pathom 73170, Thailand; piyawat.kat@mahidol.edu (P.K.); danusorn.pot@mahidol.ac.th (D.P.); rung.palakai@gmail.com (R.P.); 2Thai Health Promotion Foundation, 99/8 Ngam Duphli Alley, Thung Maha Mek, Sathon, Bangkok 10120, Thailand; niramon@thaihealth.or.th

**Keywords:** sedentary behavior, Thai population, Covid-19 pandemic, surveillance

## Abstract

Measures to contain the spread of coronavirus disease 2019 (Covid-19) imposed by governments have undoubtedly impacted on preventing its spread but may have also produced longer periods of sedentary living across all segments of society. To examine this phenomenon, this study compared the sedentary behavior (SB) of Thai adults before and during the Covid-19 pandemic. The 2019 and 2020 datasets of Thailand’s Surveillance on Physical Activity (SPA) were employed. A total of 5379 (SPA2019) and 6531 (SPA202020) persons age 18–64 years who had access to the Internet were included in the analysis. Measures imposed to contain the spread of Covid-19 infection were significantly associated with lower opportunity of Thai adults for work-related physical movement, and that increased their SB, particularly with the shift from onsite to online working platforms. Cumulative SB increased from 824 (before the pandemic) to 875 min/day during the pandemic. The odds of accumulating >13 h/day of SB was highest among females, young adults, those who completed post-secondary education, unemployed or working in the non-agriculture sector, having a chronic disease/condition, residing in an urban area, and living in a ‘higher-risk’ pandemic zone. The insignificant association of physical activity (PA) and the Fit from Home (FFH) intervention in reducing SB during the pandemic suggests that PA is not directly associated with SB, and that the FFH intervention was insufficient to prevent SB.

## 1. Introduction

The importance of sedentary behavior (SB) research is as important as studies of physical activity (PA) considering the potentially adverse effects of SB on health [1]. Studies have documented SB as one of the risk factors for cardiovascular disease (CVD) [2,3,4,5,6]. The mortality risk of CVD increases for every 2 h additional sitting time and >2 h of television watching per day [2], which could be averted by substituting SB with PA at any intensity [3]. SB was also strongly associated with poor mental health [7]. Individuals who were exposed to greater durations of screen time were more likely to report the highest level of depression symptoms and had a higher likelihood of adverse mental health outcomes [7,8,9,10,11]. Spending more time being sedentary has also been associated with a higher degree of anxiety [10,12,13].

While the Thai government has been promoting organized PA during the pandemic (e.g., Fit from Home), increased SB could not be avoided. The nationwide lockdown policy that was imposed in March 2020 had the effect of pressuring most of the Thai population to remain at home for at least three months. The closures of business and public facilities have also caused many Thais to shift from activities outside the home/neighborhood into more home-based enterprise. The ‘Working from Home’ (WFH) policy has replaced most face-to-face meetings with online video-conferencing, which has had the unintended effect of requiring more sedentary time among workers.

This study compared SB of Thai adults before and during the Covid-19 pandemic. Prevalence of SB is hypothesized to have been highest when the nationwide maximum curfew was in effect, as measured by increased minutes/day of being sedentary. This study also examined the determinants of SB during the Covid-19 pandemic among Thai adults. The results of the study should provide the government with evidence of the effect of its response to the Covid-19 on SB of Thai adults. The findings from the study also provide a baseline to continue to monitor SB patterns in order to reformulate strategies to meet the Thailand National Physical Activity Plan targets.

## 2. Materials and Methods

### 2.1. Study Population, and Sample

The 2019 and 2020 datasets from Thailand’s Surveillance on Physical Activity (SPA) survey were employed for the comparative analysis of SB before and during the period of the Thai government’s aggressive response to the Covid-19 pandemic. SPA2019 was chosen to represent SB before Covid-19, and SPA2020 represents the pandemic period. The SPA2019 used multistage stratified random sampling to select a nationally-representative sample. A total of 5379 adults aged 18–64 years who had access to the Internet were included in the analysis.

Given the restrictions on inter-personal contact to prevent transmission of Covid-19, the SPA2020 collected data using an online survey. All Thais age 18–64 who could access the Internet were defined as the population of the study. The online population was drawn by calculating the number of Internet users from the database of the Thai National Statistical Office, classified by province, as a proportion of the actual Thai population. A probability random sampling technique was applied to select participants for SPA2020 from Facebook. Facebook was selected as the platform because it provides the location of its users to enable research team members in applying multi-stages random sampling for selecting the study participants. We classified people by the area of residence (provinces) and sampled two districts in each province. In each district, we sampled Facebook open groups and invited participants by systematic random sampling. The sample was inclusive for those who have a clear sex specification on the profile page and were aged 18–64 years, yielding a total of 6531 SPA2020 cases for analysis.

### 2.2. Data Collection and Measurements

While face-to-face interviews were employed in SPA2019, an online self-administered LimeSurvey was used in SPA2020. The online survey was conducted at in a period (between March and May) relatively the same as the regular surveys to avoid bias due to environmental effects (i.e., weather, season). The Global Physical Activity Questionnaires (GPAQ) v.2 Thai version was used to measure SB in both face-to-face and online surveys. The questionnaire was validated in 2013 by using a Feel-fit accelerometer as a standard objective measure and tool validation. The correlation of cumulative minutes moderate-to-vigorous physical activity (MVPA) measured by GPAQ v.2 Thai version and Feel-fit resulted in a value of 0.809, indicating the instrument is in an acceptable level to measure the SB of the Thai population.

Sedentary behavior (SB) is defined as “any waking behavior characterized by an energy expenditure ≤1.5 metabolic equivalents (METs), while in a sitting, reclining or lying posture” [14]. We asked the respondents the following question: “On a typical day, how long do you sit, lie down, or are in a reclining position that involves only minimal movement, for example, when using a laptop or electronic device, watching TV, or using a mobile phone, excluding time spent sleeping?” We also requested the respondents to record their hourly activities (broken down into 15 min segments if there is a change of activities) for the past 24 h to ensure consistency of response. For multivariate analysis (binary logistic regression), we used 13 h per day as the cutoff point, as the Thailand National Physical Activity Plan has the target that 80% of the population are physically active and have <13 h of SB per day by 2025 [15].

Socio-demographic characteristics such as sex, age, urban/rural residence, educational attainment, and occupation were included as explanatory variables of SB. Sex was differentiated into (1) male and (2) female, whereas age was categorized into two groups: (1) young adult (18–39) and (2) middle-aged adults (40–64). Classification in age groups was made considering differences in the social roles, lifestyle and behavior between two groups of adults, particularly since middle-age adults (40+) are having multiple roles and responsibilities, enjoying little leisure, and start to experience physical decline [16].

Educational attainment was grouped into: (1) lower and primary education, (2) secondary education and (3) post-secondary education. Occupation was classified into: (1) student, (2) formal sector, (3) informal sector, (4) private enterprise, (5) agriculture, or (6) unemployed. To differentiate participants’ lifestyle and behavior driven by socioeconomic development where they are living, type of resident of the participants was defined based on administrative area: (1) urban, for those who resided in municipalities or (2) rural, for those who resided in a local administrative office.

PA and the presence of a debilitating chronic disease/condition was also observed, as those factors theoretically predict SB. A comparison of SB was also made between different epidemic zones (red-orange-green), whether the respondents received the Fit from Home (FFH) intervention provided by the Thai Ministry of Public Health (MOPH), the Thai Health Promotion Foundation (ThaiHealth) and their partners, and whether they have been adversely financially affected by the Covid-19 pandemic.

The SPA received ethical approval from the Institute for Population and Social Research of Mahidol University: COA. No. 2019/04-152 and 2020/04-190. Participants in the face-to-face interviewed indicated their agreement to be involved in the study by signing a written informed consent, whereas participants in the online survey by ticking boxes. Information on the objectives of the study and the right to join or withdraw from the research at their convenience was provided for all participants.

### 2.3. Data Analysis

Although the different methods of data collection could cause one to question the comparability of the data, several measures were undertaken in order to minimize bias. First, as the online users were mostly young adults and adults who can access the Internet, only persons aged 18–64 years in both rounds of the SPA were included in the analysis. Second, to ensure the validity of the instrument, the test-retest method was conducted with 30 individuals comparable to the sample population, and responses were compared between face-to-face and online questionnaires. Bland–Altman [17] coefficient (mean difference = 0.16, t = −0.026) and Pearson’s correlation (0.882) suggest that two methods are comparable with no significant difference. Third, to test individual items, the paired *t*-test was applied to confirm there was no significant difference between offline (face-to-face) and online responses in all items. Fourth, to control sampling bias, two groups of samples that were included and excluded in the SPA2019 were tested.

To observe differences in people’s behavior during the pandemic, data were analyzed in three different points of time: (1) before the pandemic (BC), (2) during the maximum curfew (DC) between 29 March and 2 May, and (3) after the maximum curfew was relaxed (AC) on 2 May onward. To compare cumulative minutes of SB between SPA2019 and SPA2020, the *t*-test was employed in the analysis. One-way analysis of variance (ANOVA) was used to compare SB in three different time periods. Multivariate analysis with a binary logistic regression model with maximum likelihood estimation (MLE) using SPSS software was applied to determine factors associated with SB during the Covid-19 pandemic. Autocorrelation, collinearity and multicollinearity were not detected in the model as the correlation matrix between all independent variables was below 0.65.

## 3. Results

### 3.1. Socio-Demographic Characteristics of Survey Participants

The proportion of the participants from both surveys is almost equal between sexes, but skewed toward younger adults with post-secondary education in the SPA2020. While most of SPA2019 participants were employed in the formal sector and private enterprise, about a third of SPA2020 participants worked in the informal sector. In terms of area of residence, the proportion or urban dwellers in the SPA2020 was slightly higher than in SPA2019 (66.2 versus 53.3%, respectively) (Table 1).

The government’s aggressive response to Covid-19 effectively confined most of the population to their immediate neighborhood, and that had the effect of disrupting the regular routine of physical movement for many Thais. The proportion of the population with sufficient MVPA declined from 74.6% in SPA2019 to 57.0% in SPA2020, whereas the proportion of Thais accumulating >13 h of SB per day increased from 65.9% in SPA2019 to 69.2% in SPA2020 (Table 1). The average duration of respondents in a non-sleeping, sedentary state also increased from 824 min (*SD* = 148) in SPA2019 to 875 (*SD* = 186) minutes per day in SPA2020.

### 3.2. Increased Sedentary Behavior (SB) during the Coronavirus Disease 2019 (Covid-19) Pandemic

There is no doubt that several measures to contain the Covid-19 virus, including lockdown and the WFH policy, increased the SB of the population. While, in the pre-Covid-19 period, Thai adults reported an average of 824 min of SB per day, that amount was significantly increased to 875 min during the pandemic (t = 16.582, *p*-value = 0.000) (Figure 1a). The highest SB was observed during the period of maximum curfew imposed from 29 March to 2 May, and slightly decreased after the maximum curfew was relaxed. The significance of containment measures in differing people’s behavior shown by the mean difference of SB in three different periods: before curfew (BC), during maximum curfew was imposed (DC) and after the maximum curfew was lifted (AC) and confirmed by one-way ANOVA test results at F = 133.94 with *p*-value = 0.000 (Figure 1b).

The proportion of Thai females who accumulated >13 h of SB per day is consistently higher than males in both surveys. However, a slightly different pattern was observed in SB by age group. The proportion of younger adults (18–39) with high SB was similar to their middle-age (40–64) counterparts in SPA2019, but noticeably higher in SPA2020 (Table 2). The SB by educational attainment also showed a consistent pattern in that higher SB increased with educational attainment. Higher level of SB (i.e., >13 h a day) was most frequent among students, the unemployed, and those working in the formal sector. In both pre-Covid-19 and pandemic periods, those with a chronic disease/condition had slightly higher proportion with high SB than those without a chronic condition. A high level of SB should be consistent with insufficient PA, but that was only the case in SPA2019. In SPA2020, the proportion of sample with SB >13 h a day was nearly equal between those with sufficient or insufficient PA.

### 3.3. Correlates of SB during the Covid-19 Pandemic

While the government has conducted campaigns to promote PA, some adults undergo a prolonged period of SB for various reasons, such as work-related circumstances or having to cope with a chronic disease/condition. This study found that the proportion of Thais with >13 h/day of SB increased during the pandemic period, accompanied by an increase in total sedentary time. SB was at its highest during the maximum nationwide curfew (29 March–2 May), and then declined slightly thereafter.

The non-linear regression approach using binary logistic analysis was employed to determine the factors associated with SB of the Thai population during the Covid-19 pandemic and having <13 h SB was designated as the reference. The −2LogLikelihood (−2LL) and model chi-square value (0.000) suggested that the model is explaining more of the variance in the outcome compared to the baseline model before the explanatory variables were entered. R-square value 0.052 indicated that roughly, 5.2% of the variation in the SB was caused by sex, age, education, occupation, having a chronic disease, area of resident, pandemic zone, and the exposure of FFH intervention. Autocorrelation, collinearity and multicollinearity were not detected in the model.

Females were more likely to accumulate >13 h/day of SB compared to males. The probability of having >13 h/day of SB was lower among middle-age (40–64 years) adults but higher among Thais who completed post-secondary education, worked in the informal sector, were unemployed, were a student, or had a chronic disease/condition (Table 3).

Persons who resided in the designated “highly-infectious” (red) zone were more likely to be sedentary than those is less-affected zones. Binary logistic regression analysis showed that those who lived in the orange and red zone were 1.55 and 1.28 times more likely to have >13 h/day of SB than those who lived in the green zone. Urban dwellers were also more likely to report >13 h/day of SB compared to their rural counterparts. Surprisingly, there was no effect of PA sufficiency and exposure to the FFH intervention on SB on this sample of the population. There was also no association between SB and whether or not an individual was adversely affected financially by Covid-19 (Table 3).

## 4. Discussion

With the state-ordered measures to contain the spread of Covid-19, the increase in SB could not be avoided. Total sedentary time of Thai adults rose from 824 min (*SD* = 148) before the pandemic to 875 min (*SD* = 186). Increased in SB during the Covid-19 pandemic also have been reported in various studies, ranged from 20 to 120 min in average [18,19,20]. This means, extra sedentary times collected by Thai adults during the pandemic fell at a medium level if compared to global settings. It should be noted, however, that Thai adults spent a high level of sedentary time (13.7 h) before the pandemic, mostly due to screen activities. On average, Thais spent 10 h of total screen time in 2019 [21,22]. During the pandemic, with the containment measures (i.e., travel restriction, Working from Home policy), the total amount of time Thai adults spent in front of the screen increased to 11.5 h [23].

This study also found that the highest level of SB occurred during the maximum nationwide curfew enforcement (29 March to 2 May 2020). The heightened containment measure was not only limiting individuals to travel from their home province to another, but also restricting their movement within their immediate neighborhood from 22.00 h. to 04.00 h. in the following day with a penalty of 40,000 Baht or up to 1 year of imprisonment for those who failed to comply [24].

Many studies have documented that females are less physically active and more sedentary than males [25,26,27]. Body composition, innate physical strength, and self-efficacy were often pointed to as the cause [28,29,30,31,32], apart from cultural factors (e.g., domestication, belief in fair skin as an asset) which may contribute to the higher level of SB of women in Thai society [33,34,35]. During the pandemic period, females were reported to have a higher compliance with self-isolation measures [36] and higher psychological distress [37], and that may have led them to be more house-bound and, thus, more sedentary than males.

During the pandemic, the higher SB among younger Thais was impacted by the closure of schools, increased use of distance learning and chronic use of electronic media for gaming and entertainment while confined to the home. While it is difficult to differentiate an individual’s primary motivation in accessing media, (e.g., work-related, socialization, communication, or entertainment), research found that young people more frequently use screen media than their older counterparts [38,39,40], and are more likely to use it for entertainment purposes [41,42].

In both the pre-Covid-19 and pandemic periods, urban Thais had higher SB than their rural counterparts. Most rural Thais are occupied in agricultural activities [43] that require PA in physically demanding transportation and farming. By contrast, urban Thais rely more on the motorized transportation and are employed in jobs that involve sitting in front of a computer terminal for much of the work day [44]. During the pandemic, the government’s Covid-19 containment policy was also more strictly enforced in cities, given the fact that more densely populated settlements provide a greater opportunity for easy spread of infection. Early in the Thai epidemic, many cases occurred in urban clusters, such as persons attending a nightclub or boxing stadium [45]. This study also found that SB was higher among those who lived in orange and red zones (i.e., more cases of Covid-19) which may reflect harsher containment enforcement and more individual concern about travel outside the home.

A behavioral epidemiology perspective [46] suggested that steps should be undertaken in studying the effects of the Covid-19 pandemic to people’s behavior (i.e., PA, SB). This study has established the linkage between Covid-19 and SB, developed the methods to measure SB, and identified the factors that influence SB during the pandemic. We found no effect of PA sufficiency and the FFH intervention on SB of the sample during Covid-19. The lack of a statistically significant association suggests that PA is independent of SB. An individual who engages in regular, vigorous PA could be compensating for having to spend most of their workday in sedentary activities. In addition, the FFH may have had less impact on SB since it focused on promoting home-based PA and PA guidance and techniques. Specific interventions on SB, therefore, should be developed in order to provide evidence for the government and policy makers in refining strategies in public health promotion, particularly in improving PA and reducing SB for the current and future pandemic.

While this study has provided important insights into the dynamics of SB during the extraordinary situation of Covid-19, a few limitations of the study should be acknowledged. Firstly, the self-reported measure of SB could lead to over- or under-estimation of the actual cumulative SB, as the respondent’s ability to recall their activities and its duration may have led to inaccuracy of the estimate of SB. To overcome this limitation, comparison between SB to PA and sleep data was made to ensure three behavioral components are not over 24 h. Secondly, SB of children and the elderly could not be documented and compared to the previous survey due to the limited sample size. Thirdly, although several measures have been undertaken to ensure comparability of two surveys with different data collection methods, a significant difference in the education, occupation and area of residence between two samples could not be controlled. However, stratification by region, age and sex was practiced during the sampling design to ensure representativeness of the adult population.

## 5. Conclusions

It cannot be denied that the aggressive measures imposed by the government to contain the epidemic have prevented the spread of Covid-19 in contrast to many other countries of the world which have struggled to control the outbreak. These measures also had the by-product of disrupting the daily routine of Thai adults of work-related physical movement and, thus, increased SB, particularly with the shift from onsite to online working platforms. The probability of accumulating >13 h/day of SB was the highest among females, young adults, those who attained post-secondary education, the unemployed, those working in farming, those coping with a chronic disease, those residing in an urban area, and those residing in a higher-risk epidemic zone.

The Covid-19 pandemic is affecting many dimensions of health, and most countries have focused their prevention efforts on social distancing, wearing a mask, and hand hygiene. However, less attention has been paid to the SB that confinement is causing for both adults and children. As this study did not detect any significant effect of PA and the FFH intervention in reducing SB during the pandemic, it suggests that PA is not directly associated with SB, and that the FFH intervention was insufficient to prevent SB. The results from the current study call for intensifying the SB-reduction interventions in the post-pandemic period in order to prevent adverse health outcomes due to prolonged SB. While the government should provide adequate health promotion messages or interventions on SB reduction, the workplace should encourage its employees to stand up and move more during working hours, without sacrificing productivity.

## Figures and Tables

**Figure 1 ijerph-18-04467-f001:**
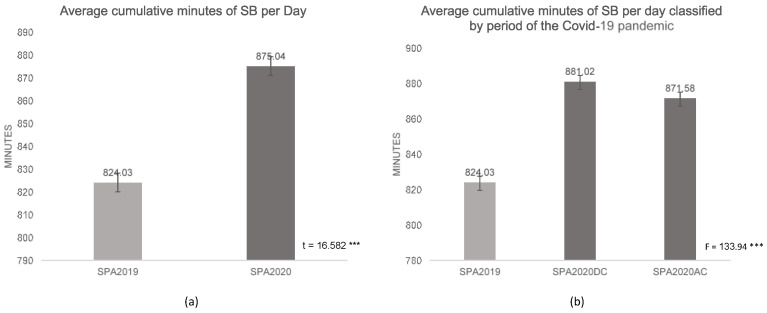
Average cumulative minutes of SB per day of the Thai population. (**a**) Average cumulative minutes of sedentary behavior (SB), (**b**) Average cumulative minutes of SB classified by the period of the Covid-19 pandemic: BC-DC-AC. BC (before curfew) March–May 2019, DC (during maximum curfew) 29 March to 2 May, and AC (after maximum curfew was relaxed). ***: significant at *p*-value 0.000. SPA: Surveillance on Physical Activity. SB: Sedentary Behavior.

**Table 1 ijerph-18-04467-t001:** Socio-demographic characteristics of survey participants.

Variable	SPA2019 (*n* = 5379)				SPA2020 (*n* = 6531)				Z	*p*-Value
			95% C.I.				95% C.I.			
	*n*	%	Lower	Upper	*n*	%	Lower	Upper		
Sex										
Male	2607	48.5	47.0	49.6	3326	50.9	49.7	52.1		
Female	2772	51.5	50.4	53.0	3205	49.1	47.9	50.3		
Age group (years)										
Young adults (18–39)	2359	43.9	42.5	45.1	4513	69.1	67.8	70.2		
Middle-age adults (40–64)	3020	56.1	54.9	57.5	2018	30.9	29.8	32.2		
Education level									1426.1	0.000
Primary and lower	1964	36.5	35.4	38.0	572	8.8	8.0	9.5		
Secondary Education	925	17.2	16.2	18.3	804	12.3	11.5	13.2		
Post-secondary Education	2490	46.3	44.8	47.4	5155	78.9	77.9	80.0		
Occupation									623.8	0.000
Student	212	4.0	3.4	4.5	423	6.6	6.0	7.2		
Formal sector	1214	22.7	21.5	23.9	1410	21.9	20.9	23.0		
Informal sector	986	18.4	17.4	19.4	2258	35.1	34.1	36.4		
Private Enterprise	1164	21.8	20.6	22.9	1342	20.9	19.8	21.9		
Agriculture	900	16.8	15.7	17.9	403	6.3	5.7	6.9		
Unemployed	871	16.3	15.4	17.3	589	9.2	8.4	9.8		
Having a debilitating chronic dis-ease/condition										
Yes	1170	21.8	20.7	22.9	1570	24.0	22.8	25.0		
No	4209	78.2	77.1	79.3	4961	76.0	75.0	77.2		
Area of residence									171.3	0.000
Urban	2866	53.3	51.9	54.6	4321	66.2	64.8	67.1		
Rural	2513	46.7	45.4	48.1	2210	33.8	32.9	35.2		
Having sufficient MVPA										
Yes	3329	74.6	72.9	75.3	3722	57.0	56.2	58.6		
No	1131	25.4	24.7	27.1	2809	43.0	41.4	43.8		
Living in a Covid-19 risk zones (as of March 2020)										
Red					2152	33.0	31.8	34.2		
Orange					3923	60.0	58.9	61.4		
Green					456	7.0	6.3	7.6		
Exposed to the ‘Fit from Home’ (FFH) campaign										
Yes					1734	26.6	25.5	27.7		
No					4797	73.4	72.3	74.5		
Adversely affected by Covid-19 pandemic										
Yes					5608	85.9	85.0	86.7		
No					923	14.1	13.3	15.0		
Having Sedentary Behaviour (Over 13 h/Day)										
Yes	3546	65.9	64.6	67.2	4472	69.2	68.1	70.3		
No	1833	34.1	32.8	35.4	1995	30.8	29.7	31.9		

Notes: SPA: Surveillance on Physical Activity. C.I.: Confidence interval. Z: Z-score. *p*-value: significance level. Formal sector includes (1) civil servants (2) politicians (3) officers (4) factory worker (5) retired civil servants. Informal sector includes (1) freelance (temporary, non-permanent workers) (2) professional athlete. Red zone: >10 confirmed positive cases. Orange zone: 1–10 confirmed positive cases. Green zone: no infection has been reported. Lockdown period: during the nationwide maximum curfew enforcement (29 March to 2 May). Relaxed curfew: after 2 May.

**Table 2 ijerph-18-04467-t002:** Characteristics of sample with >13 h of SB per day.

Variable	SPA2019						SPA2020					
	Percentage	95% C.I.		*S.D.*	*X* ^2^	*p*-Value	Percentage	95% C.I.		*S.D.*	*X* ^2^	*p*-Value
		Lower	Upper					Lower	Upper			
Overall	65.9	64.6	67.2	0.474			69.2	68.1	70.3	0.462		
Sex					71.2	0.000					6.3	0.012
Male	60.3	58.3	62.0	0.490			67.7	66.1	69.2	0.468		
Female	71.2	69.5	72.8	0.453			70.6	69.2	72.3	0.456		
Age group (years)					1.2	0.268					49.0	0.000
Young adults (18–39)	65.1	63.1	67.1	0.477			71.8	70.6	73.3	0.450		
Middle-age adults (40–64)	66.6	64.8	68.0	0.472			63.1	60.9	65.2	0.483		
Education level					73.1	0.000					68.6	0.000
Primary Education and lower	60.3	58.2	62.5	0.489			58.1	54.0	62.4	0.494		
Secondary Education	61.8	58.1	65.0	0.486			61.4	58.0	64.6	0.487		
Post-secondary Education	71.8	70.1	73.6	0.450			71.6	70.4	72.7	0.451		
Occupation					277.8	0.000					125.4	0.000
Student	75.0	69.5	81.1	0.434			77.3	73.1	81.3	0.420		
Formal sector	75.4	72.9	77.8	0.431			68.0	65.5	70.3	0.467		
Informal sector	74.4	71.8	77.1	0.436			73.8	71.8	75.5	0.440		
Private Enterprise	56.5	53.8	59.2	0.496			64.6	62.1	67.3	0.478		
Agriculture	47.8	44.4	51.2	0.500			49.2	44.6	54.5	0.501		
Unemployed	71.6	68.7	74.7	0.451			72.1	68.5	75.7	0.449		
Having a debilitating chronic dis-ease/condition					8.1	0.005					2.3	0.130
Yes	69.4	66.8	72.1	0.461			70.7	68.5	73.1	0.455		
No	65.0	63.4	66.3	0.477			68.7	67.4	70.0	0.464		
Area of residence					45.2	0.000					41.0	0.000
Urban	70.0	68.0	71.6	0.459			71.8	70.5	73.2	0.450		
Rural	61.3	59.3	63.1	0.487			64.0	62.0	66.1	0.480		
Having sufficient MVPA					616.0	0.000					5.5	0.019
Yes	56.4	54.8	57.9	0.496			68.0	68.1	70.3	0.462		
No	93.0	91.7	94.3	0.255			70.7	69.1	72.4	0.455		
Living in a Covid-19 risk zones (as of March 2020)											33.8	0.000
Red							73.6	71.6	75.6	0.442		
Orange							67.6	66.2	69.1	0.468		
Green							61.6	57.1	66.2	0.486		
Exposed to the ‘Fit from Home’ (FFH) campaign											2.4	0.118
Yes							67.4	65.4	69.9	0.468		
No							69.8	68.5	71.0	0.460		
Adversely affected by Covid-19 pandemic											0.5	0.474
Yes							69.0	67.8	70.3	0.463		
No							69.9	67.1	73.3	0.458		

Notes: SB: Sedentary Behavior. SPA: Surveillance on Physical Activity. C.I: Confidence interval. *X*^2^: Chi-square value. *p*-value: significance level. Formal sector includes (1) civil servants (2) politicians (3) officers (4) factory worker (5) retired civil servants. Informal sector includes (1) freelance (temporary, non-permanent workers) (2) professional athlete. Red zone: >10 confirmed positive cases. Orange zone: 1–10 confirmed positive cases. Green zone: no infection has been reported. Lockdown period: during maximum curfew enforcement (29 March to 2 May). Relaxed curfew: after 2 May.

**Table 3 ijerph-18-04467-t003:** Correlates of SB during Covid-19 pandemic.

Variable	Odds Ratio	*p*-Value	95% C.I. for EXP(B)	
			Lower	Upper
Sex				
Male (Ref.)				
Female	1.120	0.044	1.003	1.250
Age group (years)				
Young adults (18–39) (Ref.)				
Middle-age adults (40–64)	0.707	0.000	0.626	0.798
Education level				
Lower and primary (Ref.)				
Secondary	1.065	0.586	0.849	1.336
Post-secondary	1.532	0.000	1.267	1.854
Occupation				
Agriculture (Ref.)				
Student	2.360	0.000	1.724	3.231
Formal sector	1.745	0.000	1.382	2.205
Informal sector	2.191	0.000	1.744	2.752
Private enterprise	1.615	0.000	1.279	2.039
Unemployed	2.072	0.000	1.573	2.731
Having a debilitating chronic disease/condition				
No (Ref.)				
Yes	1.212	0.004	1.063	1.383
Area of residence				
Rural (Ref.)				
Urban	1.267	0.000	1.128	1.423
Living in a Covid-19 risk zones (as of March 2020)				
Green (Ref.)				
Orange	1.550	0.000	1.242	1.935
Red	1.284	0.019	1.043	1.581
Exposed to the ‘Fit from Home’ (FFH) campaign				
Yes (Ref.)				
No	1.087	0.184	0.961	1.229
Adversely affected by Covid-19 pandemic				
No (Ref.)				
Yes	1.027	0.752	0.872	1.209
Having sufficient MVPA				
Yes (Ref.)				
No	1.080	0.173	0.967	1.208
Constant	0.501	0.000		
df	16			
−2 Log Likelihood	7623.676			
Cox and Snell R^2^	0.037			
Nagelkerke R^2^	0.052			
Model Chi-square	241.296	0.000		
Number of observations	6531			

Notes: SB: Sedentary Behavior. C.I.: Confidence interval. *p*-value: significance level. EXP (B): exponentiation of the B coefficient. Ref.: reference category. -2Log Likelihood: natural logarithm of the likelihood. df: degree of freedom. R^2^: R squared. Formal sector includes (1) civil servants (2) politicians (3) officers (4) factory worker (5) retired civil servants. Informal sector includes (1) freelance (temporary, non-permanent workers) (2) professional athlete. Red zone: >10 confirmed positive cases. Orange zone: 1–10 confirmed positive cases. Green zone: no infection has been reported. Lockdown period: during maximum curfew enforcement (29 March to 2 May). Relaxed curfew: after 2 May.

## Data Availability

SPA data is available in TPAK repository, https://tpak.or.th/?p=4151 (accessed on 7 August 2020).

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
