# Peer review of "The Effect of Containment Measures during the Covid-19 Pandemic to Sedentary Behavior of Thai Adults: Evidence from Thailand’s Surveillance on Physical Activity 2019–2020"

_ijerph, 2021, doi:10.3390/ijerph18094467_

Round 1

Reviewer 1 Report

In the abstract line 10 remove "the" and pluralize "government"

In the abstract introduce the abbreviation for PA.

line 70 explain more on how you are using facebook and how this won't later create a convenience sample of more sedentary people. 

line 95, flesh out more of what each of these groups are for and how you define rural and urban. Are there no suburbs in thailand?

Line 111 this is not the same as doing a stat test for things being the same, make sure you do this as well.

results, report data as mean +/- SD.

table 1 run stats for sig difference in 2019 to 2020 data for education, occupation, area of residence.

line 138 explain what freelance means. 

figures need error bars and need a y axis.

table 3 or results in general a part and partial correlation between metrics would be quite useful to see.

discussion is good, but explain more the limitations to the timing and how this might have influenced behaviors. How shortly after lockdown were people surveyed? Has there been a follow up since to see if behaviors have reverted?

Author Response

Responses to reviewer 1 attached. 

Reviewer 2 Report

This paper examined sedentary behavior of Thia adults before and during the COVID-19 pandemic. Overall, these observations contain practical relevance among Thais and those interested in how the ongoing pandemic influenced PA and overall health and wellness. 

Some comments of mine: 

Abstract:

I assume PA is physical activity, but I do not see this denoted in the abstract. I do, however, see Surveillance of Physical Activity (SPA). 

Line 16/17 was this association significant? If so, please include "significant"

Introduction: 

Please add a reference to the sentence in lines 28-29 and 33-34. 

Methods: 

line 75-76 I think there is an extra "the" in there, should it be "..relatively same period.."?

Please add a reference at the end of line 87-88. 

Out of curiosity...were additional analyses performed stratified by menopausal status?

Please consider the use of sex as opposed to gender. In light of this, I would suggest including a note indicating something along the lines of "The analyses were also performed stratified by sex (e.g., males and females)." Please be cognizant that gender does not adequately convey male vs female. 

Results and Discussion: 

Overall, I thought these sections were very well articulated and easy to follow. The authors did a great job here. 

Author Response

Responses to reviewer 2 attached.

Reviewer 3 Report

The study was designed to compared sedentariness of Thai adults before and during the Covid-19 pandemic. Prevalence of sedentariness was assumed to be high during the restricted period of activities. The introduction is well-written and is very clear.

Although the number of participants is high, an estimate of the power is required as it does not mention the numbers of the whole population. The SPA program must have calculated this measure. Note I am not asking the SPA sample size, but how representative of Thailand’s population. Besides, the older population may have created a bias in the analysis as some of them may be on their first years of retirement. Please, clarify the age they retire or the impact of these subjects on your data.

Line   34 – Please, specify the direction and magnitude of the relationship, which is negative, I suppose.

Please, explain the rationale for breaking up the group into two halves according to age. You have age in a continuous scale, which is compatible with your regression analysis. This will provide variations regarding age instead of age groups with a large range. Also, please, provide some evidence regarding the reliability of your questionnaires. Were they validated?

There is no mention to ethical committee’s approval, which is a requirement for collecting data of any population. Please, clarify and provide the approval reference.

The bias is present in any condition. The correlation is not a good way to determine reliability. Please, consider other more robust approaches (e.g., Bland Altman’s coefficient). You mention that your t-test did not show differences but did not provide the actual numbers. Remember that 0.051 would be the same as 0.99 – and we know the strength of the relationship is somehow very different.

The regression equation is poorly explained as there are several parameters that were omitted and are relevant for one to understand how authors conducted the analysis. No mention of the software was provided. In SPSS provides many parameters that are mandatory for reporting such sort of analysis. No comments are made in terms of collinearity control, type of regression approach, method used for entering the independent variables.

MVPA? Please, provide a description for abbreviations and acronyms.

In many places authors indicate “an increase”, which I am not confident as there is no statistical treatment to support such speculation. As far as I understand, authors are inferring from the magnitude of the numbers. Please, clarify, especially because Table 1 does not provide much information beyond the main means (i.e., no SDs, interquartile or other measures of variation are informed). I strongly suggest table 1 to be fully revised because no units are provided, and readers are left on their own to deduct what units are being presented. Finally, the captions mention information not provided in methods and include some strange information not disclosed before (i.e., zone colors – how was it included in the analysis?). It confuses the reviewer, and I am sure will also puzzle readers.

I am also confused on how authors concluded 824 differs from 875 (SB minutes); please add the proper statistical values (t values are meaningless without other relevant information). I suppose the F-test refers to an ANOVA with repeated measures. Is that correct? Please, expand description and inform the statistical requirements for applying such approach (e.g., data normality).

I am very confused by the lack of clarity in the analysis. For instance, authors compare middle-age but there is no mention to such group.

Table 1 and 2 may be merged.

You are not discussing vigorous activities. PA was defined as…”???”. So, discussing female’s behavior based on these arguments does not seem to be very useful.

Please, address the following statement, (“as the respondent’s ability to recall their activities and its dura- 261 tion may have led to inaccuracy of the estimate of SB.”), as your data should have controlled for that. This is why I asked for a better approaches to ensure your data is reliable (please, avoid correlational approaches).

Author Response

Responses to reviewer 3 attached.

Round 2

Reviewer 3 Report

I congratulate authors for implementing and expanding several parts os the manuscript that has experienced a good improvement.

Although most of changes were implemented, I still insist authors revise the stats section as most of the analysis are not properly described within that section, i.e., readers discover that an ANOVA was applied in the results. The full description of all procedures must be clearer in that particular section. 

In addition, I asked for several other information regarding you regression equation (model, collinearity, and so on). Please, state that such approaches were cautiously performed. For instance, authors inform the reviewer they controlled for collinearity, but no information regarding this was provided. What was the critical cut0ff limit to include/exclude variables in the model. I stress that this information MUST be present and clear in the proper section (not disclosed only to the reviewer). Please, be more informative. 

Author Response

Responses to reviewer 3.

Reviewer’s comment:

I congratulate authors for implementing and expanding several parts os the manuscript that has experienced a good improvement.

Although most of changes were implemented, I still insist authors revise the stats section as most of the analysis are not properly described within that section, i.e., readers discover that an ANOVA was applied in the results. The full description of all procedures must be clearer in that particular section.

In addition, I asked for several other information regarding you regression equation (model, collinearity, and so on). Please, state that such approaches were cautiously performed. For instance, authors inform the reviewer they controlled for collinearity, but no information regarding this was provided. What was the critical cut0ff limit to include/exclude variables in the model. I stress that this information MUST be present and clear in the proper section (not disclosed only to the reviewer). Please, be more informative.

Response:

We have added relevant information on statistic model and its assumption on method section, data analysis part as follows:

To compare cumulative minutes of SB between SPA2019 and SPA2020, the t-test was employed in the analysis. One-way ANOVA was used to compare SB in three different time periods. Multivariate analysis with binary logistic regression model with maximum likelihood estimation (MLE) using SPSS software was applied to determine factors associated with SB during the Covid-19 pandemic. Autocorrelation, collinearity and multicollinearity were not detected in the model as correlation matrix between all independent variables were below 0.65.